# Research Progress in Enzymatic Synthesis of Vitamin E Ester Derivatives

Zhiqiang Zou [1], Lingmei Dai [1], Dehua Liu [1,2] and Wei Du [1,2,*]

[1] Key Laboratory for Industrial Biocatalysis, Department of Chemical Engineering, Tsinghua University, Ministry of Education, Beijing 100084, China; zouzq19@mails.tsinghua.edu.cn (Z.Z.); dailm@tsinghua.edu.cn (L.D.); dhliu@tsinghua.edu.cn (D.L.)

[2] Tsinghua Innovation Center in Dongguan, Dongguan 523808, China

* Correspondence: duwei@tsinghua.edu.cn

**Abstract:** Vitamin E is easily oxidized by light, air, oxidizing agents and heat, limiting its application in many ways. Compared to vitamin E, vitamin E ester derivatives exhibit improved stability and a stronger antioxidant capacity, and even gain new biological functions. In recent years, enzymatic synthesis of vitamin E ester derivatives has received increasing attention due to its environmental friendliness, high catalytic efficiency, and inherent selectivity. This paper reviews the related progress of lipase-mediated preparation of vitamin E ester derivatives. The function of different vitamin E ester derivatives, and the main factors influencing the enzymatic acylation process, including enzyme species, acyl donor and acceptor, reaction media and water activity, are summarized in this paper. Finally, the perspective of lipase-catalyzed synthesis of vitamin E ester derivatives is also discussed.

**Keywords:** antioxidants; derivatives; lipase; vitamin E; vitamin E ester

## 1. Introduction

Vitamin E (VE) is a group of eight liposoluble compounds, including four tocopherols and four tocotrienols. They are all composed of benzodihydropyran rings and 2-position phytogroups (Table 1). Tocopherol has three chiral carbon atoms, and there are three isolated double bonds on the phytogroups of tocotrienol [1]. Vegetable oil extracted from oil-bearing seeds and nuts is the most abundant source of natural vitamin E. The total vitamin E content in different oils varies from thousands ppm to tens ppm, and contains a variety of tocopherols and tocotrienols [2]. Chemically synthesized vitamin E is mainly a racemic alpha-tocopherol by using trimethylhydroquinone and racemic isophytol [3].

Natural antioxidant vitamin E is more suitable for food additives, health care, cosmetics and other industries compared to synthetic antioxidants such as terbutyl hydroquinone (TBHQ), butyl hydroxytoluene (BHT), butyl hydroxyanisole (BHA), and propyl gallate, which may form toxic or carcinogenic compounds after degradation [4,5].

Vitamin E is easily oxidized and is unstable to light and singlet oxygen. The antioxidant property of VE is reflected either in its termination of the free radical chain reaction (Figure 1a), or in the way that itself can be oxidized and lose its antioxidant property (Figure 1b). Herein, when vitamin E is extracted from natural sources or is synthesized by chemical methods, it easily loses its original antioxidant function when added to products due to its own oxidation during the process of production, transportation, and storage. Therefore, the modification on vitamin E to improve its stability is significant in promoting its wider application [5].

It is well recognized that vitamin E ester derivatives have a wide range of applications in pharmaceuticals, cosmetics, daily chemicals, and other fields. At present, the most popular vitamin E ester products in the market are vitamin E acetate, vitamin E succinate, vitamin E linoleic acid ester, and vitamin E niacin ester. Vitamin E acetate has been demonstrated to be effective in dealing with inflammation elimination, periodontal

disease prevention, ulcer healing, blood microcirculation enhancement of gingival, and anti-aging [6]. Vitamin E succinate can effectively inhibit the growth of tumor cells without affecting the proliferation of normal cells [7]. Vitamin E linoleate has excellent skin moisturizing properties and is mainly used in the cosmetic industry as an additive in advanced cosmetics [8].

**Table 1.** Structure of vitamin E.

| Structural Formula | Type | $R_1$ | $R_2$ | $R_3$ | $R_4$ | |
|---|---|---|---|---|---|---|
| | $\alpha$ | $CH_3$ | $CH_3$ | $CH_3$ | | |
| | $\beta$ | $CH_3$ | H | $CH_3$ | Tocopherol | Tocotrienol |
| | $\gamma$ | H | $CH_3$ | $CH_3$ | | |
| | $\delta$ | H | H | $CH_3$ | | |

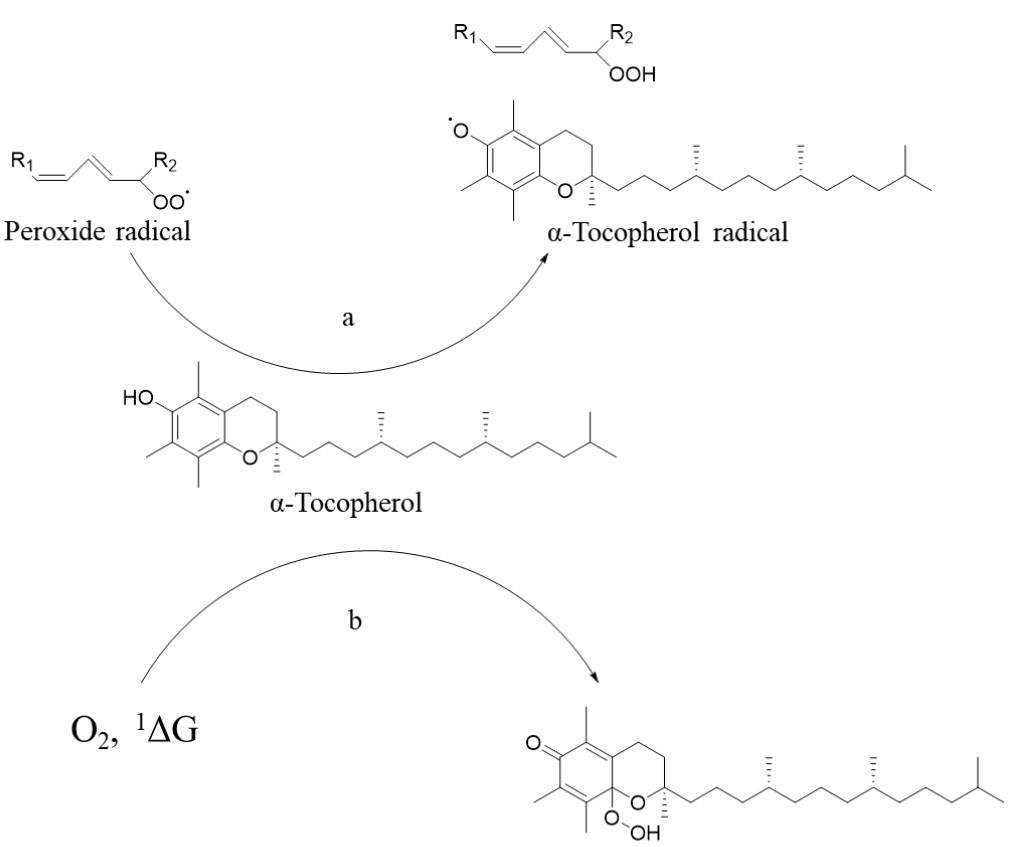

**Figure 1.** (**a**) VE terminated free radical chain reaction, (**b**) VE quenched singlet oxygen itself oxidized.

Vitamin E ester derivatives are mainly synthesized by chemical and enzymatic methods (Figure 2). The chemical method mainly adopts a Lewis acid and an organic base as the catalysts. Chen Dan used nano-silica and immobilized 4-dimethylaminopyridine to catalyze the reaction of $\alpha$-tocopherol with succinic anhydride in *n*-hexane-acetone mixed solvent to produce alpha-tocopherol succinate [9]. Chen Xuebing used pyridine to catalyze the reaction of acetic anhydride and $\alpha$-tocopherol to produce $\alpha$-tocopherol acetate in a solvent-free system, obtaining a 99.4% conversion rate [10]. Hu Chuanrong used triethylamine to catalyze the reaction of $\alpha$-tocopherol and succinic anhydride in *n*-hexane with a 93.91% conversion rate [11]. Despite the fact that chemical synthesis of vitamin E ester derivatives are extensively studied, enzymatic catalysis offers a great perspective due to its milder reaction conditions, simpler product separation, and environmental friendliness.

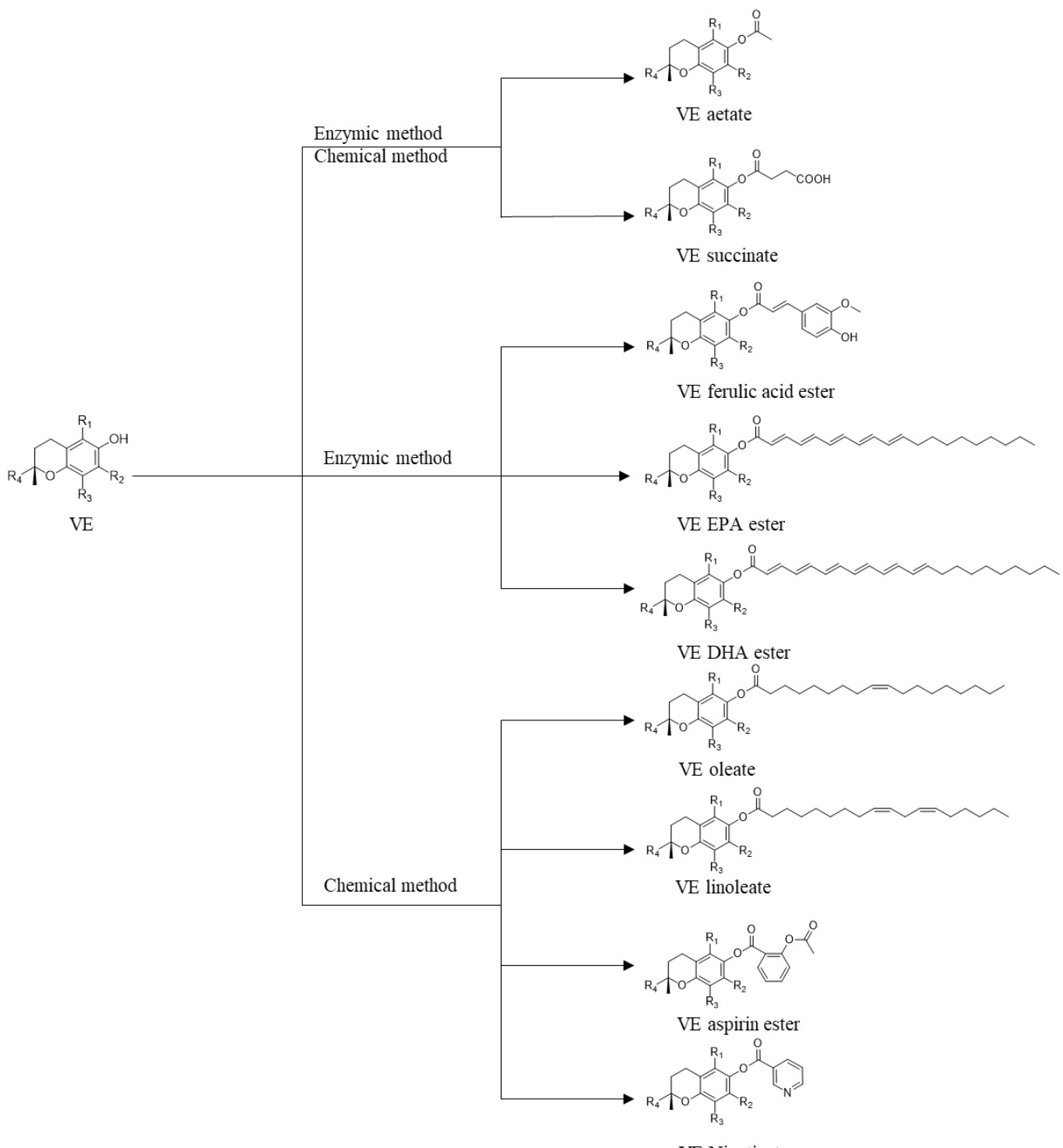

**Figure 2.** Various VE ester derivatives and their preparation methods.

## 2. Main Function of VE Ester Derivatives

Generally speaking, vitamin E esters are more stable and have a longer shelf life than vitamin E in the presence of light and oxygen. Studies have shown that the equimolar quantity of vitamin E ester has better potency than corresponding vitamin E since vitamin E has serious oxidative destruction before being absorbed by the small intestinal wall in mice [12–14]. Vitamin E ester is hydrolyzed by esterase to free vitamin E, and, with blood circulation, free vitamin E spreads to various tissues to perform this function [15].

Vitamin E esters still retain free radical scavenging capacity. Free radicals generated by UV-induced melatonin interacted with vitamin E and vitamin E ester derivatives, including vitamin E acetate, vitamin E linoleic acid ester, and vitamin E succinate, and their free radical scavenging ability was tested [16,17]. The results showed that vitamin E linoleic acid ester had the inhibitory effect of 25.6% at the mass concentration of 1%, while vitamin E succinic acid ester had the inhibitory effect of 37.6% at the mass concentration of 0.1% [17]. The DPPH free radical scavenging experiment showed that vitamin E acetate still had the same free radical scavenging ability as vitamin E [18]. The above experiments indicated that the esters of vitamin E did not lose their antioxidant capacity, but even had a stronger antioxidant capacity than vitamin E. When used as an effective ingredient in cosmetics, vitamin E ester can effectively prevent the skin from oxidative damage [19]. In the experiment of ultraviolet-induced skin tumors in mice, both vitamin E succinate and vitamin E can effectively inhibit tumor formation [20].

Some vitamin E ester derivatives were found to have new biological functions. Vitamin E succinate was found to be capable of inhibiting the growth of estrogen receptor-negative human breast cancer cells and inducing their apoptosis and death [21]. Vitamin E nicotinate had important applications in lowering blood pressure, lowering body fat, and inhibiting virus proliferation [15]. Experiments in mice showed that the concentration of endogenous vitamin E nicotinate was negatively correlated with heart failure [22].

## 3. Research Status of Enzymatic Preparation of Vitamin E Ester Derivatives

Lipase-mediated synthesis of vitamin E acetate, vitamin E succinate, vitamin E DHA/EPA ester, and vitamin E ferulic acid ester are widely studied (Figure 3).

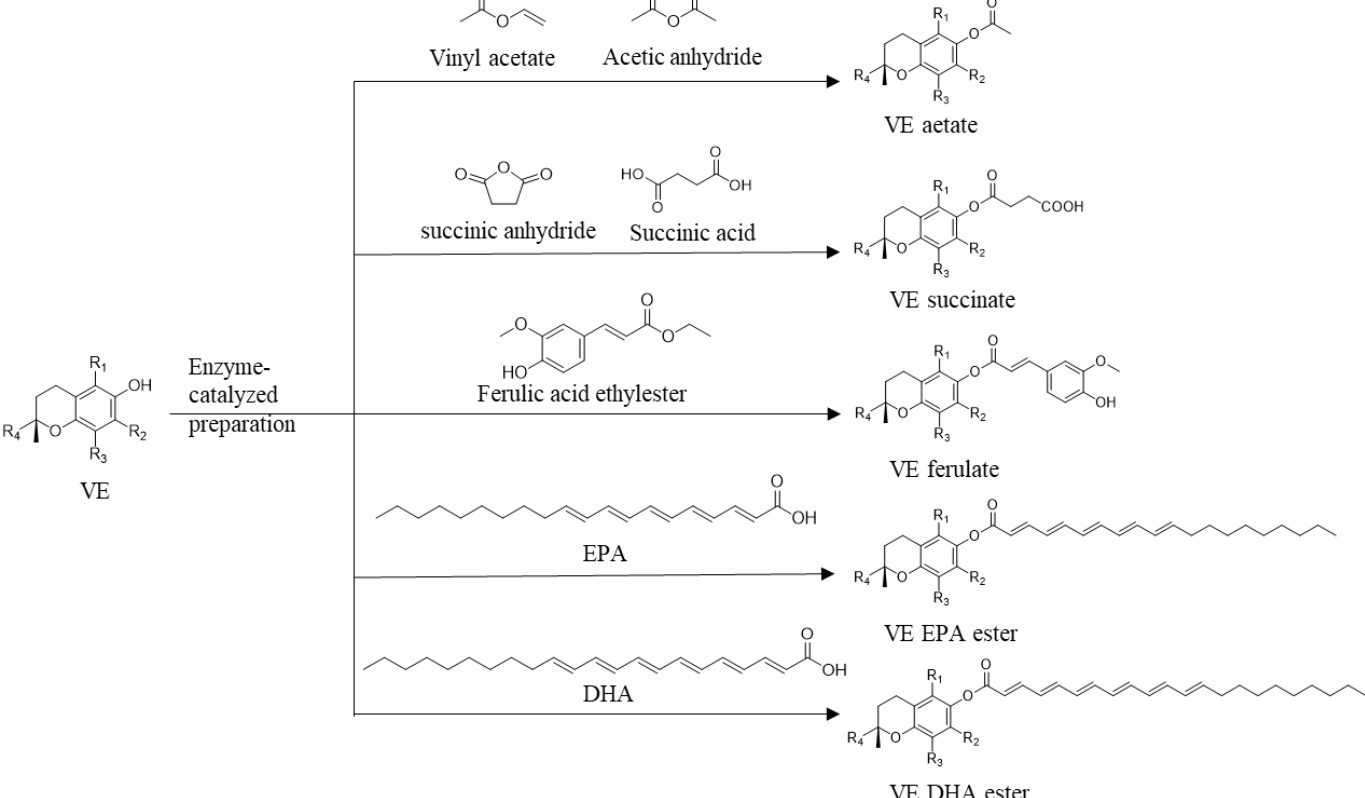

**Figure 3.** Enzymatic preparation of VE esters with various acyl donors.

### 3.1. VE Acetate

Vitamin E acetate is one of the most extensively studied VE ester derivatives. Pamela Torres et al. reported the transesterification of vinyl acetate with α-tocopherol catalyzed by Novozym 435 in mixed solvent of *n*-hexane and 2-methyl-2-butanol, with a conversion rate of 60% achieved [23]. Hao Yuechuo et al. realized the high-efficiency expression of CRL1 in *Pichia pastoris* and catalyzed acetic anhydride and α-tocopherol esterification in a solvent-free system, with a conversion rate of 97% obtained [24]. Gong et al. carried out a comparative study on the esterification of α-tocopherol catalyzed by self-made immobilized lipase in organic solvent and solvent-free system, respectively [25]. The conversion rate was above 85% in petroleum ether solvent and 95% in the solvent-free system.

### 3.2. VE Succinate

Vitamin E succinate is widely researched due to its medicinal value. Yin et al. used Novozym 435 to catalyze the synthesis of α-tocopherol succinate from succinic anhydride and α-tocopherol; the conversion rate was 94.4% [26]. Jiang et al. catalyzed the reaction of succinic anhydride and α-tocopherol to produce α-tocopherol succinate in DMSO with *Candida rugosa* lipase (CRL) as the catalyst and a yield of 46.95% was obtained [27]. Hu Y. et al. used sol-gel material to immobilize CRL for the synthesis of f α-tocopherol succinic acid ester [28]. They found that the enzymatic activity of the sol-gel immobilized enzyme was increased 6.7-fold compared to the free enzyme, while the enzymatic activity of the olive oil activated sol-gel immobilized enzyme was increased 1.43-fold compared to the pre-activation (the micromolar number of α-tocopherol converted into succinic acid ester per gram of enzyme per hour was defined as the enzyme activity unit). Jiaojiao X. et al. immobilized CRL to catalyze the formation of vitamin E succinate; the conversion rate was 62.58% [29]. Zhao Junbo et al. used Lip400 to catalyze the reaction of α-tocopherol with succinic acid in dichloromethane; the yield of VE succinate was 80% [30].

### 3.3. Other VE Esters

It was reported that introducing functional acyl donors could enable vitamin E esters novel functions. Zu et al. successfully prepared α-tocopherol DHA/EPA ester by introducing polyunsaturated fatty acids (PUFA) into α-tocopherol with the catalysis of Lipozyme RM IM [31]. Xin et al. used CRL to catalyze the transesterification reaction between ethyl ferulate and α-tocopherol in toluene with a yield of 72.3% [32]. Xin et al. adopted Novozym 435 to catalyze the transesterification of ethyl ferulate and α-tocopherol in the solvent-free system; the conversion rate was 35.4% [33].

## 4. Main Factors Influencing Enzymatic Synthesis of Vitamin E Ester Derivatives

### 4.1. Enzyme

*Candida antarctica* lipase *B* (CALB), *Rhizomucor miehei* lipase (RML), *C. rugosa* lipase (CRL), and other lipases have been proven to catalyze the synthesis of vitamin E ester derivatives [23–31]. The properties of the enzyme itself, the immobilized carriers, and further modification on enzyme molecules have varied effects on VE ester synthesis.

The catalytic performance of different enzymes is shown in Table 2. Pamela Torres et al. studied the transesterification of vinyl acetate with α-tocopherol and found that only Novozym 435 could catalyze the reaction [23]. They thought that CALB had a deeper substrate binding site than other enzymes, which was key to the lipase-mediated transesterification. Studies also indicated that Lipozyme RM IM could catalyze the esterification of α-tocopherol with DHA/EPA [31], and that CRL could catalyze the esterification of α-tocopherol with succinic acid [27]. Jürgen Pleiss et al. compared the geometric structures and active sites of different lipases (Figure 4) [34]. They found that the hydrophobic region in the active site of RML was larger than that of CALB. The geometric structure and characteristics of active sites was thought to play a key role in catalyzing the synthesis of different VE esters.

**Table 2.** Catalytic performance of different lipases in catalyzing the synthesis of VE esters.

| Enzyme | Types of VE Esters | Conversion (%) | Ref. |
|---|---|---|---|
| Novozym 435 | α-tocopherol acetate | 60 | [23] |
| | α-tocopheryl ferulate | 35.4 | [33] |
| CRL | α-tocopherol acetate | 97 | [24] |
| | α-tocopherol succinate | 46.95 | [27] |
| | α-tocopheryl ferulate | 72.3 | [32] |
| succinic anhydride modified Novozym 435 | α-tocopherol succinate | 94.4 | [26] |
| Nanogel immobilized CRl | α-tocopherol succinate | 62.58 | [29] |
| Lipozyme RM IM | α-tocopherol DHA/EPA ester | 77.65 | [31] |
| Immobilized Candida sp. 99–125 | α-tocopherol acetate | 39.37 | [25] |
| Lip400 | α-tocopherol succinate | 86 | [35] |

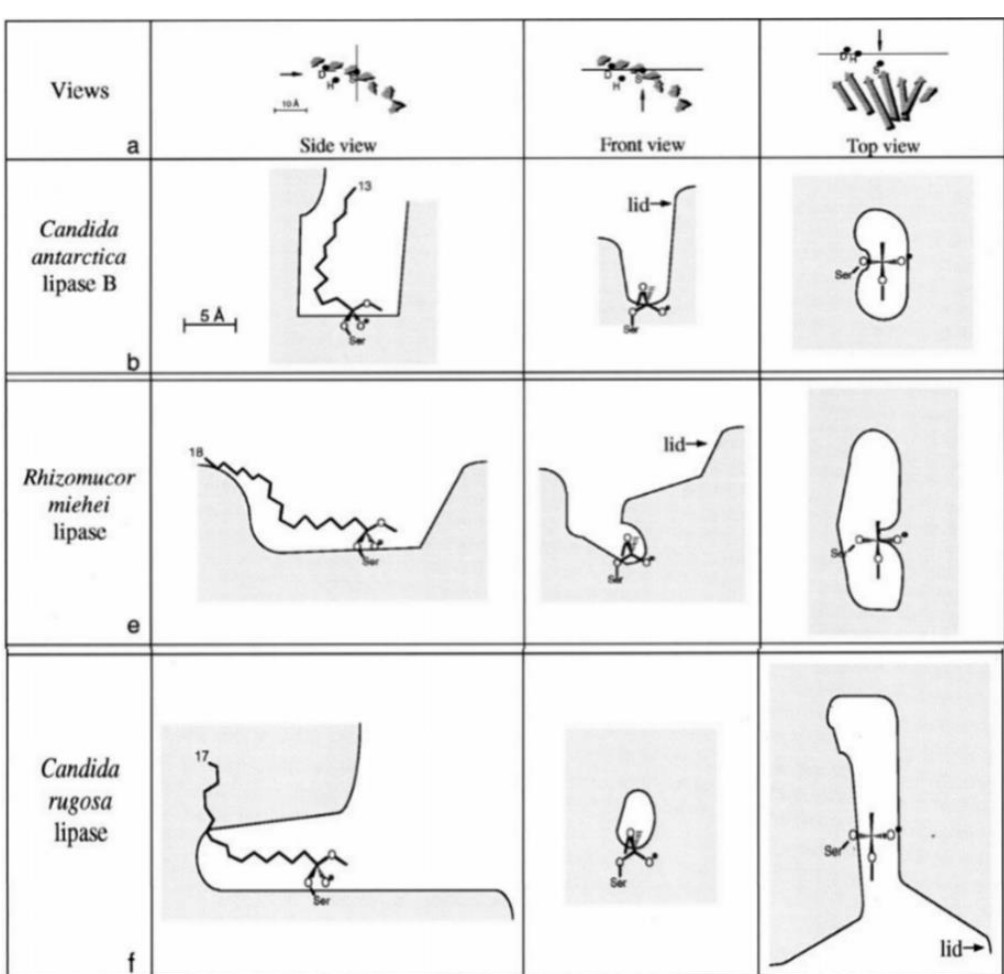

**Figure 4.** Trigrams of lipase active sites in CRLB, RML and CRL [34]. Reprinted from Chemistry and Physics of Lipids, Vol. 93, Jürgen Pleiss, Markus Fischer, Rolf D Schmid, Anatomy of lipase binding sites: the scissile fatty acid binding site, Pages No. 67–80, Copyright 1998, with permission from Elsevier.

Immobilized carriers have varied influence on the enzymatic catalysis during VE ester's preparation. Pamela Torres et al. explored the effect of different immobilized materials and found that the catalytic effect of CALB immobilized by polypropylene

material with larger pore size and pore volume was more effective than other materials [23]. Jiaojiao X. used immobilized CRL to catalyze the formation of vitamin E succinate and found that CRL immobilized by polyacrylamide nanogels had a layer-by-layer stacking structure, a larger specific surface area and a larger pore volume [29]. Hu Y. immobilized CRL with sol-gel material; they found that the ratio of hydrophobic and hydrophilic silane precursors not only affected the micro-phase separation, but also influenced the immobilization efficiency as well as the specific enzyme activity [28].

Using specific reagents to modify the surface of enzyme molecules can enhance the catalytic effect of the enzyme. It was found that modification on the primary amino groups of enzyme molecules by using organic acids could greatly enhance the activity and thermal stability of the enzyme [36,37]. Yin Chunhua modified Novozym 435 with succinic anhydride; the enzyme's thermal stability as well as the catalytic activity both improved. Hu Y. revealed that modification on enzyme molecules by using surfactants could open the hydrophobic lid of the enzyme's active site with an increased activity achieved [28].

### 4.2. Reaction Medium

Since vitamin E is highly viscous and water-insoluble, most studies related to enzymatic synthesis of VE esters are mainly carried in an organic solvent system.

Pamela Torres explored the effect of solvents during Novozym 435-catalyzed transesterification of vinyl acetate with $\alpha$-tocopherol [23]. A solvent mixture of 2-methyl-2-butanol and *n*-hexane was used. It was found that the conversion rate increased with the increase in the *n*-hexane addition ratio, but the conversion rate in pure *n*-hexane solvent was lower than that in pure 2-methyl-2-butanol solvent. They speculated that the addition of the *n*-hexane reduced the polarity of the solvent and increased the enzyme activity, but that the presence of a certain amount of the 2-methyl-2-butanol promoted the dissolution of vinyl acetate. In the reaction of succinic anhydride with $\alpha$-tocopherol catalyzed by Novozym 435, Yin Chunhua found that, in a single solvent system, the yield was higher in DMF and DMSO [26]. DMF and DMSO could dissolve water-soluble and fat-soluble substrates well, but strong solvent polarity would deprive the necessary water on the surface of the enzyme molecule and affect the conformation of the enzyme molecule, resulting in lower enzyme activity. In order to find a balance between the solubility of the substrates and a relatively high enzyme activity, they used DMSO and tert-butanol to prepare a mixed solvent. The highest yield was obtained in the system when the volume ratio of DMSO and tert-butanol was 3:2. Zu Guoren used Lipozyme RM IM to catalyze the reaction of fish oil with $\alpha$-tocopherol to prepare polyunsaturated fatty acid esters, and investigated the effects of different organic solvents on the reaction [31]. It was found that the conversion rate in the *n*-hexane solvent was the highest. Xin Jiaying explored the effect of different reaction mediums on the transesterification of ethyl ferulate and $\alpha$-tocopherol, catalyzed by Novozym 435 [33]. They found that there was almost no reaction in tert-butanol, and that the conversion rate in toluene was 17.7%. The logP of solvent was thought to be the main parameter influencing the reaction during lipase-mediated preparation of VE esters. The related results are summarized in Table 3.

A few researchers have investigated the enzymatic synthesis of VE esters in the solvent-free system. When preparing $\alpha$-tocopherol acetate, Gong et al. conducted a series of optimizations in organic solvent and solvent-free system. It was found that the highest yield of $\alpha$-tocopherol in the organic solvent system was 85%, while in the solvent-free system the yield could reach 95% [25]. Xin Jiaying studied the synthesis of VE ferulic acid ester and found that the highest conversion rate of ethyl ferulate in the solvent-free system was 23.8%, while the highest conversion rate in the organic solvent was only 17.7% [33].

Despite of the fact that extensive research has been carried out to explore enzyme-catalyzed synthesis of VE esters in organic solvents, the use of organic solvents always causes problems of environmental pollution and increased difficulty in product separation. The solvent-free system or other green reaction systems needs to be explored further.

**Table 3.** VE esters synthesized by enzymatic method in different solvents.

| Solvents | | logP | α-tocopheryl Acetate Yield (%) | | α-tocopherol Succinate Yield (%) | | | α-tocopherol DHA/EPA EsterYield (%) | α-tocopheryl Ferulate Yield (%) |
|---|---|---|---|---|---|---|---|---|---|
| logP < 0 | DMSO | −1.35 | / | / | 78.12 | 28.37 | 31.01 | / | / |
| | DMF | −1.0 | / | / | 53.30 | 10.30 | 14.62 | / | / |
| | Acetonitrile | −0.33 | / | / | 37.54 | 1.35 | 1.87 | / | / |
| | Ethyl alcohol | −0.24 | / | 1.82 | / | / | / | / | / |
| | Acetone | −0.23 | / | 3.77 | 4.43 | / | / | / | / |
| 0 < logP≤ 2 | Tetrahydrofuran | 0.49 | / | 2.48 | / | / | / | / | / |
| | Ethyl acetate | 0.68 | / | / | / | 1.01 | 1.71 | / | / |
| | T-butanol | 0.80 | / | / | 23.71 | / | / | / | 0 |
| | 2-Methyl-2-butanol | 1.17 | 5 | / | / | / | / | / | / |
| | T-amyl alcohol | 1.3 | / | / | / | 0.97 | 1.40 | / | / |
| | Benzene | 2.0 | / | 4.28 | / | / | / | / | / |
| | Chloroform | 2.0 | / | / | / | / | / | 47.00 | / |
| logP > 2 | Toluene | 2.5 | / | / | 17.89 | 1.04 | 1.53 | / | 17.70 |
| | Cyclohexane | 3.2 | / | 15.80 | / | / | / | / | / |
| | Petroleum ether | ~3.5 | / | 49.00 | 8.85 | / | / | 49.68 | / |
| | N-hexane | 3.5 | 7 | 39.37 | 10.47 | 0 | 0 | 61.00 | 6.90 |
| | N-heptane | 4.0 | / | 34.82 | / | / | / | / | / |
| | Isooctane | 4.5 | / | / | / | / | / | / | 3.10 |
| Ref. | | | [23] | [25] | [26] | [27] | [29] | [31] | [33] |

### 4.3. Acyl Donors and Acceptors

As the most active vitamin E isomer, α-tocopherol is widely studied. In addition to α-tocopherol, oils also contain large numbers of other tocopherols and tocotrienols. Although other tocopherol isomers usually exhibit lower biological activity in organisms, some of them exhibit stronger antioxidant properties in vitro antioxidant studies [38]. Therefore, the acylation of other vitamin E isomers is of great significance for the full utilization of natural vitamin E. During the preparation of different VE esters, a variety of acyl donors can be selected, such as long-chain or short-chain fatty acids, anhydrides, fatty acid esters, ethyl ferulate, and so on. The selection of suitable acyl donors to acylate different VE isomers will contribute greatly to the full utilization of natural VE.

Different isomers of vitamin E have different effects on the acylation reaction. Pamela Torres et al. used α-tocopherol and δ-tocopherol, two different acyl receptors, to react with vinyl acetate catalyzed by Novozym 435 [23]. It was found that δ-tocopherol was higher than α-tocopherol in terms of reaction rate and final conversion, which might be related to its less steric hindrance.

Various acyl donors also bring different reaction results. For the preparation of VE acetate, the commonly used acyl donors are vinyl acetate and acetic anhydride. Pamela Torres reported the transesterification of vinyl acetate and α-tocopherol catalyzed by Novozym 435 [23]. α-Tocopherol reacted with vinyl acetate for 18 days to achieve a 60% conversion rate. Hao Yuechuo reported the reaction of acetic anhydride and α-tocopherol catalyzed by CRL1 with the conversion rate of 97% obtained [24]. For the preparation of VE succinate, Yin used Novozym 435 to catalyze succinic anhydride and α-tocopherol to synthesize α-tocopherol succinate [26]. After a 48 h reaction, the conversion rate reached 94.4%. Zhao Junbo used Lip400 to catalyze the reaction of α-tocopherol with succinic acid in dichloromethane solvent to prepare VE succinate with a 80% yield after a 6 h reaction [30].

*4.4. Water Activity*

Water activity has complicated the influence on the reaction of enzymatic synthesis of vitamin E ester derivatives. Water molecules directly or indirectly maintain the three-dimensional conformations necessary for the catalytic activity of enzyme molecules through hydrogen bonds, hydrophobic interactions, van der Waals forces, etc. Moreover, during the esterification for vitamin E ester preparation, by-product water is generated and the control of water content during the whole process also needs to be taking into consideration.

The optimal water activity required by different enzymes often varies. Anna E.V. Petersson constructed an accurate water activity control system for the kettle reactor, and used Novozym 435 and immobilized CRL to catalyze the esterification of hexadecanol with palmitic acid [39]. It was found that, for Novozym 435, a faster catalytic rate and a higher conversion rate could be achieved at lower water activity, while the opposite was observed for the case of CRL. The optimal water activity of Novozym 435 and CRL was found to be 0.02 and 0.5, respectively. Xin Jiaying et al. explored the effect of water activity during the synthesis of α-tocopherol ferulic acid ester which was close to 25%, while, when the water activity was over 0.23, the average conversion rate was only 5%.

## 5. Purification of VE Ester Derivatives

Solvent crystallization, urea complexation and molecular distillation are widely used for the purification of VE esters [25].

C.D. Robenson proposed a solvent crystallization method for the separation of VE acetate [40]. The experimental system was pyridine-catalyzed esterification of α-tocopherol with acetic anhydride. At the end of the reaction, pyridine, anhydride, and acetic acid were washed with 5% hydrochloric acid. The organic phase was distilled and the fractions at 130–180 °C were collected as crystallized raw materials. The residual acetic acid was removed by crystallization in the methanol solution at −30 °C, and then the needle-like crystallization of α-tocopherol acetate was obtained by crystallization in the methyl formate solution at −30 °C. For urea complexation, VE esters dissolved in organic solvents and crystallized with the added urea, and then other components such as fatty acids and fatty acid esters were removed, and VE esters with high purity could be obtained [41]. Isolation of VE esters generally require multi-stage molecular distillation. On the basis of primary molecular distillation, the temperature of the heating wall increased and the heavy phase obtained after primary molecular distillation was subjected to a second molecular distillation to separate it from vitamin E. Similarly, tertiary molecular distillation further increased the heating wall temperature to separate vitamin E esters [42].

## 6. Conclusions and Foresight

As a natural antioxidant, vitamin E is widely used in food, feed, and cosmetics because of its no potential toxic effect. The conversion of vitamin E into vitamin E ester derivatives can not only improve the stability of vitamin E itself, but also enhance its antioxidant activity. Even the introduction of different acyl donors will obtain some new biological functions. Acylation of vitamin E plays an important role in promoting the sustainable development of the vitamin E industry.

There are many kinds of vitamin E esters prepared by this chemical method. At present, the industrial preparation of VE esters adopts a chemical catalytic process. Chemical synthesis of VE ester mainly uses a Lewis acid, an organic base, and other catalysts. With the increasing awareness of global environmental protection, the enzymatic preparation of VE esters has great potential.

**Author Contributions:** Z.Z.: Literature collection & manuscript writing; L.D.: Literature collection and analysis; D.L.: Content design; W.D.: Content design & revision. All authors have read and agreed to the published version of the manuscript.

**Funding:** This research received the fund from from Dongguan Science & Technology Bureau (Innovative R&D Team Leadership of Dongguan City, 201536000100033).

**Conflicts of Interest:** The authors declare no conflict of interest.

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
