# Peer review of "Research Progress in Enzymatic Synthesis of Vitamin E Ester Derivatives"

_catalysts, doi:10.3390/catal11060739_

Round 1

Reviewer 1 Report

The review paper submitted by Wei Du and co-authors is very instructive description of state of the art of the preparation of ester derivatives of Vitamin E group of compounds modulated by their enzymatic synthesis.

Vitamin E is one of the most useful antioxidant applied comprehensibly in medicinal, food, cosmetics and other applications. Their instability could be improved by esterification which could be performed by classical chemical methods, combination of chemical and enzymatic protocols or enzymatic methods. The presented review is dealing with recently developed enzymatic methodologies.

The article is well organized and brings all necessary information useful not only for those familiar with the matter at academic community, but also for interested industry. The introduction to the reviewed matter is very instructive, while the concussion remarks bring critical observations and proposals for future development. Overall, the submitted paper is good example of how the any special matter should be presented in a review compilation of the facts, their critical evaluation and future perspective.

I suggest the publication of the present submission as it is.

Author Response

Thanks.

Reviewer 2 Report

The paper is understandable and written clearly and, from my perspective, deserves the scientific public's attention. However, I suggest the following corrections to improve the paper quality:

  • It seems this is a shorter review article, maybe a mini-review. Nevertheless, it contains (only) 42 references, and thus my suggestion is to increase the number of references as much as possible. It seems this is a shorter review article, maybe a mini-review. Nevertheless, it contains (only) 42 references, and thus my suggestion is to increase the number of references as much as possible.
  • - The authors describe the enzymatic lipase-mediated synthesis of vitamin E esters. It is known from the literature that some lipases (e.g., from P. aeruginosa ATCC 27853), isolated from an extreme environment, are resistant to extreme conditions such as organic solvents etc. Could this type of lipase be used for the synthesis of vitamin E-based compounds? Is there literature on this topic?
  • - English is generally good and clear, but I noticed some errors. Thus, I recommend professional English editing.

Author Response

  • It seems this is a shorter review article, maybe a mini-review. Nevertheless, it contains (only) 42 references, and thus my suggestion is to increase the number of references as much as possible. It seems this is a shorter review article, maybe a mini-review. Nevertheless, it contains (only) 42 references, and thus my suggestion is to increase the number of references as much as possible
  • ----We included almost all the possible related references and it is ok for us to publish this as a sort of mini-review.
  • - The authors describe the enzymatic lipase-mediated synthesis of vitamin E esters. It is known from the literature that some lipases (e.g., from P. aeruginosa ATCC 27853), isolated from an extreme environment, are resistant to extreme conditions such as organic solvents etc. Could this type of lipase be used for the synthesis of vitamin E-based compounds? Is there literature on this topic?
  • ---there is no related report and this lipase can be tried for future research study.
  • - English is generally good and clear, but I noticed some errors. Thus, I recommend professional English editin
  • ---modified.

Reviewer 3 Report

This manuscript reviewed the enzymes catalyzed synthesis of esters of vitamin E.  

Although overall my evaluation is positive, I think that the manuscript can be improved first of all from the point of view of clarity, scientific rigorousness and language.  

Below are some identified minor inaccuracies and some critical points that should not be ignored:

- p.2, line 3 of the second paragraph under Fig.1: “4-dimethoxypyridine” should be replaced probably with  “4-dimethylaminopyridine”

-

line 5 of the same paragraph: “succinic” should be replaced probably with “succinate”

- in the same paragraph: “Chen Xuebing used pyridine to catalyze the reaction of acetic anhydride and α-tocopherol to produce α-tocopherol acetate in solvent-free system and obtained 99.4% esterification rate”. It is not clear what means: “99.4% esterification rate”? It seems that here “esterification rate’ is used to denote «degree of esterification”. Although this term is used in some other publications, I think that in order to avoid misunderstanding it is desirable to provide definition of the term used. Especially since this term is used repeatedly in the article.

-It seems that throughout the paper, the term “esterification rate” is used by authors in the meaning of “degree of conversion in %”. The same is true for the term “conversion rate, %”.

p.3, Item 2, line 2: “It is found that vitamin E ester is more stable and is not easily oxidized.”

This is general statement (and refers not to particular ester, but to esters in general), therefore, the plural should be used.

p.3: “equimolar vitamin E ester”?? possibly  “equimolar quantity of vitamin E ester”?

 More precise  use of terms throughout the paper is advised

p.4, item 3, first paragraph “ Despite of the fact that many vitamin E ester derivatives are successfully synthesized by chemical approaches, lipase-mediated preparation are mainly related to vitamin E ac-etate, vitamin E succinate, vitamin E DHA/EPA ester and vitamin E ferulic acid ester are studied” – correct the phrase.

p.5, item 3.3, first paragraph: “It was found that introducing some functional acyl donors could enable vitamin E ester derivatives having dual functions”  – correct the phrase.

p.6. “In this way, the explanation of Pamela Torres et al. for the experimental phenomenon is controversial, at least not because vitamin E cannot bind to the enzyme catalytic site.” The phrase is difficult to understand.

Some other identified places, needed correction are highlighted in yellow in the attached manuscript.

I think that the paper may be published in Catalysts after consideration the above points.

Author Response

Response

- p.2, line 3 of the second paragraph under Fig.1: “4-dimethoxypyridine” should be replaced probably with “4-dimethylaminopyridine”

-Yes, replaced.

line 5 of the same paragraph: “succinic” should be replaced probably with “succinate”

-Yes, replaced

- in the same paragraph: “Chen Xuebing used pyridine to catalyze the reaction of acetic anhydride and α-tocopherol to produce α-tocopherol acetate in solvent-free system and obtained 99.4% esterification rate”. It is not clear what means: “99.4% esterification rate”? It seems that here “esterification rate’ is used to denote «degree of esterification”. Although this term is used in some other publications, I think that in order to avoid misunderstanding it is desirable to provide definition of the term used. Especially since this term is used repeatedly in the article.

---- changed to conversion rate.

-It seems that throughout the paper, the term “esterification rate” is used by authors in the meaning of “degree of conversion in %”. The same is true for the term “conversion rate, %”.

---- yes.changed to conversion rate.

p.3, Item 2, line 2: “It is found that vitamin E ester is more stable and is not easily oxidized.”

This is general statement (and refers not to particular ester, but to esters in general), therefore, the plural should be used.

--yes.

p.3: “equimolar vitamin E ester”?? possibly  “equimolar quantity of vitamin E ester”?

More precise use of terms throughout the paper is advised

--yes, checked and modified.

p.4, item 3, first paragraph “ Despite of the fact that many vitamin E ester derivatives are successfully synthesized by chemical approaches, lipase-mediated preparation are mainly related to vitamin E ac-etate, vitamin E succinate, vitamin E DHA/EPA ester and vitamin E ferulic acid ester are studied” – correct the phrase.

---Yes. Changed to Lipase-mediated synthesis of vitamin E acetate, vitamin E succinate, vitamin E DHA/EPA ester and vitamin E ferulic acid ester are widely studied.

p.5, item 3.3, first paragraph: “It was found that introducing some functional acyl donors could enable vitamin E ester derivatives having dual functions”  – correct the phrase.

---yes. Changed to “It was reported that introducing functional acyl donors could enable vitamin E esters novel functions.

p.6. “In this way, the explanation of Pamela Torres et al. for the experimental phenomenon is controversial, at least not because vitamin E cannot bind to the enzyme catalytic site.” The phrase is difficult to understand.

---Modified as “More study needs to be carried out to explore the related mechanism.”

Some other identified places, needed correction are highlighted in yellow in the attached manuscript.

--corrected.

Round 2

Reviewer 2 Report

The authors have provided appropriate explanations for my suggestions. Thus, this article could be published.

In my opinion, English might be more polished.

Author Response

Yes. checked and modified.

Reviewer 3 Report

Authors did not respond properly on the use of terms “esterification rate, %”, or “conversion rate, %”.  It seems that these terms were used by authors non correctly throughout the paper with the meaning of “degree of conversion” or “degree of esterification”. Conversion in chemistry describes the ratio reactant/product and it can be defined in %. But it is quite unclear what means “rate of conversion in %”?  Rate of conversion can denotes the velocity with which some conversion is achieved.

I think that in order to avoid misunderstanding, authors should either use term “conversion, %” instead of “conversion, rate %” or provide clear definitions of the terms used.

Author Response

modified.